# Proteome Map of Pea (*Pisum sativum* L.) Embryos Containing Different Amounts of Residual Chlorophylls

**DOI:** 10.3390/ijms19124066

**Published:** 2018-12-15

**Authors:** Tatiana Mamontova, Elena Lukasheva, Gregory Mavropolo-Stolyarenko, Carsten Proksch, Tatiana Bilova, Ahyoung Kim, Vladimir Babakov, Tatiana Grishina, Wolfgang Hoehenwarter, Sergei Medvedev, Galina Smolikova, Andrej Frolov

**Affiliations:** 1Department of Bioorganic Chemistry, Leibniz Institute of Plant Biochemistry, 06120 Halle (Saale), Germany; mamontova-bio@mail.ru (T.M.); bilova.tatiana@gmail.com (T.B.); ariyong1002@gmail.com (A.K.); 2Department of Biochemistry, St. Petersburg State University, St. Petersburg 199178, Russia; elena_lukasheva@mail.ru (E.L.); gm2124@mail.ru (G.M.-S.); tgrishina@mail.ru (T.G.); 3Proteome Analytics, Leibniz Institute of Plant Biochemistry, Weinberg 3, 06120 Halle (Saale), Germany; Carsten.Proksch@ipb-halle.de (C.P.); Wolfgang.Hoehenwarter@ipb-halle.de (W.H.); 4Department of Plant Physiology and Biochemistry, St. Petersburg State University, St. Petersburg 199034, Russia; s.medvedev@spbu.ru; 5Research Institute of Hygiene, Occupational Pathology, and Human Ecology, Federal Medicobiological Agency, 188663 Kapitolovo, Russia; vbabakov@gmail.com

**Keywords:** chlorophylls, LC-MS-based proteomics, pea (*Pisum sativum* L.), proteome functional annotation, proteome map, seeds, seed proteomics

## Abstract

Due to low culturing costs and high seed protein contents, legumes represent the main global source of food protein. Pea (*Pisum sativum* L.) is one of the major legume crops, impacting both animal feed and human nutrition. Therefore, the quality of pea seeds needs to be ensured in the context of sustainable crop production and nutritional efficiency. Apparently, changes in seed protein patterns might directly affect both of these aspects. Thus, here, we address the pea seed proteome in detail and provide, to the best of our knowledge, the most comprehensive annotation of the functions and intracellular localization of pea seed proteins. To address possible intercultivar differences, we compared seed proteomes of yellow- and green-seeded pea cultivars in a comprehensive case study. The analysis revealed totally 1938 and 1989 nonredundant proteins, respectively. Only 35 and 44 proteins, respectively, could be additionally identified after protamine sulfate precipitation (PSP), potentially indicating the high efficiency of our experimental workflow. Totally 981 protein groups were assigned to 34 functional classes, which were to a large extent differentially represented in yellow and green seeds. Closer analysis of these differences by processing of the data in KEGG and String databases revealed their possible relation to a higher metabolic status and reduced longevity of green seeds.

## 1. Introduction

Legumes represent the most prominent source of food protein, and their importance is increasing with the growing global population [1]. Indeed, these crops are tolerant to environmental stressors, cheap to culture, and rich in seed protein (typically about 25% of fresh seed weight) [2,3]. Among the cultured legumes, pea (*Pisum sativum* L.) is the most widely spread pulse crop in Europe, where it serves as a protein food supplement for monogastric animals [4]. Therefore, quality of pea seeds is important from the aspects of both sustainable crop production and high nutritional efficiency. Obviously, changes in protein composition of seeds directly affect their agricultural and nutritional value [5]. 

The first pea seed protein map was reported at the end of the last decade; it was based on the two-dimensional gel electrophoresis (2D-GE) and mass spectrometric (MS) identification of visualized electrophoretic zones (spots) and contained 156 proteins [6]. The majority of the identified polypeptides were storage proteins (convicilins, vicilins, and legumins), which strongly dominate the seed proteome and can serve as seed protein quality markers [7]. Removal of these highly abundant storage proteins by extraction with aqueous (aq.) isopropanol-containing solutions [8] or by precipitation in presence of aq. 0.01–0.1% (*w*/*v*) protamine sulfate [9] could increase coverage of the seed proteome. Alternatively, as was shown for soybean (*Glycine max*), seed storage globulins (glycinin and β-conglycenin) can be effectively removed by 10 mmol/L CaCl_2_ [10]. Low-abundance proteins can be selectively enriched by means of the combinational peptide ligand libraries technology [11]. However, despite the efficiency of these techniques, in combination with the gel-based proteomics approach, the numbers of identified proteins never exceed several hundreds.

Because of this, implementation of liquid chromatography (LC)-MS-based strategy is desired to get a deeper insight into the seed proteome [12]. However, as detergents, conventionally used for solubilization of protein isolates [5], dramatically affect efficiency of electrospray ionization (ESI) [13], it is difficult to find a compromise between completeness of protein reconstitution and sensitivity of MS analysis. In this context, the introduction of degradable detergents into proteomic practice helped to overcome this caveat [14]. Thus, commercially available detergents such as RapiGest™ and Anionic Acid-Labile Surfactant II (AALS II) gave a deeper insight into the proteomes of young barley [15] and developing oilseed rape [16] seeds. 

When considering the pea seed proteome, it is necessary to remember that cultivars can differ essentially by their metabolic background. For example, in the seeds of green-seeded cultivars, chlorophylls are not quantitatively destroyed after completion of seed maturation, and the seeds of such plants preserve green color in their mature state [17]. This phenomenon is underlain by deficiency of one or several chlorophyll catabolic enzymes (CCEs), for example, chlorophyll *b* reductase, 7-hydroxymethyl chlorophyll *a* reductase, Mg^2+^-dechelatase, pheophytinase, pheophorbide *a* oxygenase, and reductase of red chlorophyll catabolite (RCC) [17]. Thus, the presence of chlorophylls in *sgr* (*stay-green*) mutants is mostly attributed to the damage of *SGR* genes, prospectively encoding the enzymes involved in chlorophyll degradation and/or disassembly of chlorophyll-protein complexes [18]. Remarkably, green seeds are characterized with higher amenability to stressors (e.g., associated with accelerated ageing [19,20]) that might be related to their higher oxidative status [21]. Adaptation to these metabolic changes might result in essential alterations in seed proteomes. These events, potentially affecting seed nutritional properties, have not been addressed so far.

Therefore, here, we provide, to the best of our knowledge, the most complete map of the pea seed proteome using sample prefractionation and LC-MS-based shotgun proteomics. We address the differences in embryo proteomes of yellow- and green-seeded pea cultivars and discuss the function and localization profiles of seed proteins in the context of possible differences in their response to varying environmental conditions.

## 2. Results

### 2.1. Analysis of Physiological and Biochemical Parameters of Seed Quality

As green seeds contain residual chlorophylls, and are, therefore, potentially more prone to development of oxidative stress [19,21], we addressed physiological and biochemical parameters, giving access to oxidative status of the embryos and accompanying changes in lipid peroxidation levels, membrane integrity, status of antioxidant defense, and functional activity of photosynthetic apparatus. Accordingly, the cultivar-specific differences in seed germination kinetics and electrolyte conductivity, as well as the contents of photosynthetic pigments, hydrogen peroxide and lipid peroxidation products, were determined. The yellow seeds of the cultivar Millennium and the green ones of the cultivar Gloriosa clearly differed in the contents of photosynthetic pigments. Thus, chlorophylls were detected solely in green seeds, and the contents of carotenoids were approximately 25% higher in these seeds (*t*-test: *p* = 5.41 × 10^−9^), in comparison to the yellow ones (Figure 1A and Appendix A). Both pea cultivars demonstrated 100% seed viability, although the seeds of the cultivar Millennium germinated faster (Figure 1B). 

Moreover, the percentage of normally developed seedlings on day ten was 13% higher for yellow seeds, although this difference was not significant (Figure 1C). Conductivity testing did not reveal differences in electrolyte leakage between yellow and green seeds (*t*-test: *p* = 0.07, Figure 1D), whereas the green seeds demonstrated statistically significant approximately two- and fourfold higher levels of lipid peroxidation products and H_2_O_2_ (*t*-test: *p* = 0.0058 and 0.0002, respectively, Figure 1E,F).

### 2.2. Protein Isolation and Tryptic Digestion

To ensure efficient extraction of seed proteins and the maximal coverage of the pea seed proteome, we decided to use phenol-based protein extraction (Figure 2), thereby resulting dry protein isolates could be reconstituted in shotgun buffer containing at least 0.15% AALS. Protein determination revealed the extraction yields were in the range of 39.6–124.1 mg/g fresh weight (Appendix A). Assay precision was determined by SDS-PAGE loading 5 µg of protein (Appendix A); the overall lane densities were 1.4 × 10^4^ ± 4.5 × 10^2^ arbitrary units (AU, RSD = 3.06%). The signal patterns observed in the electrophoregrams were similar between lanes and pea cultivars (Appendix A). Tryptic digestion of proteins was considered to be complete, as the bands of major pea storage proteins, such as legumin (α- and β- subunits, ∼40 kDa and ∼20 kDa, correspondingly), vicilin (subunits of ∼29 kDa, ∼35 kDa, and ∼47 kDa), and convicilin (subunit of ∼71 kDa), could not be detected (Appendix A), assuming a staining sensitivity better than 30 ng [22] and a legumin content of at least 80% of total seeds proteins [23].

### 2.3. Depletion of Storage Seed Proteins by Protamine Sulfate

To ensure sufficient efficiency of the PSP procedure in the presence of AALS, we diluted the protein samples 10-fold, to arrive at a final AALS concentration of 0.015% (*w/v*). Additionally, we applied the highest concentration, tested by the Kim’s group (0.07%), although the authors reported 0.05% (*w/v*) protamine sulfate as a sufficient concentration, for efficient depletion of seed storage proteins. Interestingly, the protein recoveries were higher for yellow seed embryos (71.3–84.0 mg/g fresh weight, Appendix A), than for the green ones (56.2–61.0 mg/g fresh weight). However, the recoveries after protamine sulfate depletion were slightly higher for green than for yellow seed embryos (3.7–4.2 vs. 2.9–3.8 mg/g fresh weight, respectively). Accordingly, the depletion efficiency was slightly higher for the yellow than for the green seed embryos (95.4–96.0 vs. 92.6–94.0%, respectively, Appendix A). In agreement with this observation, electrophoregrams of depleted protein extracts were clearly different for yellow and green seeds (Appendix A), although this was not the case for nondepleted samples (Appendix A). SDS-PAGE revealed depletion of the major seed storage proteins (Appendix A), which was, however, incomplete (Appendix A). Tryptic digestion of depleted samples was complete, that is, no proteins were detected in the corresponding electrophoregram (Appendix A).

### 2.4. Annotation of Pea Seed Proteins

To verify the applicability of our combined database for annotation of pea seed proteins, we manually interpreted MS/MS spectra surpassing dual FDR thresholds (strict 0.01 and relaxed 0.05), but identified with the lowest XCorrs in both yellow and green seed embryo preparations. This procedure revealed reliable identification of low-scoring peptides, and this was valid for all three proteomes comprising the database; the sequences of corresponding individual peptides could be confirmed by the numbers of b and y fragments, sufficient for their unambiguous identification (Figure 3). In total, 9162 peptides (7923 and 8292 in Millennium and Gloriosa seeds, respectively, Figure 4A, Appendix A) were identified with the FDR of 0.05. On the basis of these identifications, 8769 possible proteins could be annotated (7821 and 8134 in yellow and green seeds, respectively, Figure 4B, 0.05 protein FDR threshold), which represented 2195 nonredundant proteins, or so-called protein groups (1938 and 1989 in Millennium and Gloriosa seeds, respectively, Figure 4C). Precipitation of high-abundance proteins with 0.07% (*w/v*) protamine sulfate (PS) resulted in identification of 2399 and 2286 tryptic peptides in yellow and green seed embryos, respectively (totally 2974, Figure 4A), however, only 44 and 85 peptides were unique for PS-treated extracts of yellow and green seeds. These unique peptides, identified specifically in yellow and green seeds, gave access to 11 and 20 protein groups, represented by 24 and 42 proteins, respectively (Figure 4B,C). Further, 24 protein groups, represented by 84 proteins, were in common to yellow and green seeds, although identified solely in corresponding PS fractions. Removal of the high-abundance proteins was effective, as can be judged by high numbers of proteins and protein groups, observed exclusively in nondepleted samples (Figure 4B,C). Indeed, 1152 and 1196 proteins were removed by the PS treatment from the extracts of Millennium and Gloriosa seeds, respectively (1377 in total), that accounted for 8% and 10% of the totally identified protein groups.

### 2.5. Functional Annotation of Seed Proteins and Prediction of Their Cellular Localization

Functional annotation resulted in assignment of 981 nonredundant proteins (875 and 900 proteins, isolated from yellow and green seed embryos, respectively) to one or more of 34 functional bins, whereas 1269 polypeptides remained unassigned (Figure 5A, Appendix A). Among the assigned proteins, 926 and 380 could be annotated to functional bins in nondepleted and PSP-depleted fractions, respectively (Figure 5B,C). The bin with the most entries (274 and 275 entries for yellow and green seeds, respectively), was nucleotide metabolism (bin #23), although it included nucleotide binding proteins and ATPases (Appendix A). Bins related to photosynthesis, metal handling, miscellaneous enzymes, RNA, DNA, and protein metabolism (#1, 15, 26–29) were also large (Figure 5A). For most of the functional groups, the numbers of proteins assigned to them were similar in yellow and green seed embryos. However, the number of entries in the groups of metal-binding proteins and enzymes of amino acid metabolism were higher in green seed embryos, whereas nucleotide metabolism pathways and miscellaneous enzyme families were more represented in yellow seeds. The results of localization prediction were in agreement with the observed functional patterns, although only minimal differences between yellow and green seeds were observed (Figure 6, Appendix A).

Thus, in yellow seed embryos, representation of cytosol and plasma membrane was slightly higher (20.4% and 12.6%, respectively), whereas the nuclear fraction accounted for 1.5% less, in comparison with green seeds (Figure 6A). In green seeds, the largest fraction (approximately 46.7%) was represented by nuclear proteins, whereas the majority of the other proteins were localized to cytosol, mitochondria, and plasma membrane (19.3%, 13.7%, and 11.9%, respectively, Figure 6B). Remarkably, depletion of high-abundance products did not essentially affect this distribution (Appendix A).

To address specific metabolic features of yellow and green seeds, we considered the proteins, identified solely in one of the cultivars. Despite the general similarity of protein profiles (Figure 7A, Appendix A), miscellaneous enzyme families were better represented in yellow seeds, whereas the pathways of RNA and amino acid metabolism were more prevalent in green ones. Interestingly, green seeds contained more proteins, which could be localized to the nucleus and endoplasmic reticulum (55% and 2.7%, respectively, Figure 7B), whereas cytosol and cytoskeleton were represented more in yellow seed embryos (15% and 2%, respectively, Figure 7C).

Among 251 proteins, unique for green seed embryos (Appendix A), 237 could be annotated by way of homology to *A. thaliana* proteins and analyzed with the String database. Thereby, nine structural/functional and seven coexpression protein interaction networks could be annotated (Figure 8).

Finally, we addressed the question of how application of the protamine sulfate precipitation (PSP) affects functional profiles and protein localization. For this purpose, we compared the proteins identified only in nondepleted fractions (i.e., precipitation in the presence of protamine sulfate) and those found only after depletion of the highly abundant proteins. As can be seen from Figure 9A, precipitation of high-abundance proteins (representing mostly nucleotide, protein, amino acid, and metal-related metabolism, Appendix A) led to the identification of several other proteins annotated to protein, nucleic acid, and nucleotide metabolism. Accordingly, the supernatants obtained after supplementation with protamine sulfate were dominated by nuclear proteins, whereas the depleted proteins exhibited a higher percentage of cytosolic, mitochondrial, and plasma membrane polypeptides (Figure 9B,C).

## 3. Discussion

### 3.1. Protein Extraction and Depletion of Highly Abundant Proteome Fraction

Similarly to our earlier study [16], a high variability of protein yields (39.6–124.1 mg/g fresh weight, Appendix A) was observed. This phenomenon can be explained by biochemical heterogeneity of seeds [24,25]. Approximately 50% higher recoveries, observed for green seeds, were in agreement with a higher number of proteins, identified in this fraction (Figure 4). On the other hand, precise estimates of protein abundance (RSD = 3%) were in line with our other previous work [26], and thus allowed a comparative, label-free proteomics approach of the heterogeneous seed proteomes [22].

Due to a high abundance of major seed storage proteins [9] and a high amenability of DDA-based shotgun proteomics measurements to a so-called under-sampling effect [27], poor identification rates can be expected for low-abundance proteins. This issue can be expected to be the case when analyzing seed protein digests, as the seed proteome is dominated by several highly abundant protein families–vicilins, convicilins, 11S globulins (legumins), and 2S albumins (PA1 and PA2) [28]. Among the analytical solutions, available for selective removal of storage proteins [29], we selected the PSP method due to its high precision (CV < 12%) and, generally, in contrast to most of the other published techniques, quantitative nature [30]. Indeed, in our hands, this approach allowed extensive proteome coverage, that is, only a low number of proteins could be additionally identified after PSP (only 1.7% of the total 2195 proteins, Figure 4). Interestingly, PSP yielded mostly regulatory proteins, represented by transcription factors and regulators (GO terms “DNA binding” and “RNA binding”), kinases, transport proteins, and enzymes of energy metabolism (Appendix A). The variability of structural domains of these mostly short and polar polypeptides was much lower in PSP-treated samples (Appendix A).

### 3.2. Annotation of the Pea Seed Proteome

To date, studies of the legume seed proteome mostly relied on gel-based techniques (typically 2D-GE and MALDI-TOF(/TOF)-MS), yielding up to several hundreds of protein identifications now available [31,32]. Sample prefractionation [33] and isoelectrofocusing (variation of pH gradients) [31,34] do not dramatically improve the situation. In this context, due to the better resolution of RP-HPLC, an LC-MS-based approach seemed promising [14]. Indeed, the most representative LC-MS-based study to date of Min et al., performed with soybean seeds, identified 1626 nonredundant proteins [35], which exceeds the best outcomes from gel-based proteomics surveys by at least twofold [29]. However, it accounts for only a half of the proteins annotated here (Figure 4), although the authors relied on the same prefractionation strategy and instrumentation. The possible reasons for this fact might be (i) a longer nanoHPLC column and shallower gradient, (ii) a more efficient digestion protocol, and (iii) a more representative sequence database, used here. 

Longer analysis times with shallow gradients might allow better separation of the complex peptide mixtures [36] and a larger number of DDA cycles per run. Our digestion protocol relies on Anionic Acid-Labile Surfactant (AALS, Progenta), which ensures quantitative digestion of all proteins in a sample, and can be completely destroyed and removed upon proteolysis [37]. Recently, we optimized this procedure for the total protein fraction, obtained by phenolic extraction from various plant species and tissues [16,27]. Similarly to the procedure based on the RapiGest detergent (Waters) [15], this protocol is straightforward and does not include steps, at which losses of recovery and/or precision would be expected to occur (e.g., ultrafiltration [38] or preseparation by PAGE [39]). Finally, our sequence database, containing protein entries of three legumes related to pea, already proved to be a reliable tool for the identification of low-abundance post-translationally modified (glycated) tryptic peptides (known to yield complex fragmentation [40]) in common bean [41]. The database contained exclusively reviewed (i.e., based on transcriptome and proteome data) entries. As reviewed sequence information is currently scarce for *P. sativum*, we assumed this approach to be more appropriate than use of a genomic database. The results of manual interpretation of low-scoring, statistically significant PSMs (Figure 3) suggested reliable annotation of peptide sequences.

### 3.3. Functional Annotation of Pea Seed Proteins

Functions of pea seed embryo proteins were successfully annotated by in-house-designed processing pipeline. As can be judged by spectral counts [42] for corresponding unique peptides, storage proteins were the most abundant in proteomes of both yellow and green seeds, and their pattern of legumins, convicilins, vicilins, and glycinins corresponded well with the observation of Bourgeois et al. [6]. The other major functional classes were represented by the polypeptides involved in metabolism of proteins and RNA, photosynthesis, metal handling, redox metabolism, and general metabolism enzyme families (Figure 5). Based on the number of functionally assigned proteins (Appendix A), representation of protein metabolism in pea seeds was comparable to soybeans [39,43]. However, the metal handling, RNA metabolism, and redox groups were much more abundant here in comparison to those studies. This can be explained by deeper coverage of the seed proteome, (threefold compared to Han et al. [39]), although the incomplete functional assignment in our study makes the difference less striking.

### 3.4. Features of Embryonic Proteome, Related to Seed Vigor: Impact of Residual Chlorophylls

As pea seeds belong to the orthodox type and acquire desiccation tolerance during development [44], their ability to sustain a prolonged dehydration is the most important feature underlying their high vigor (i.e., the properties determining the potential for rapid, uniform emergence and development of normal seedlings [45]). During desiccation, stability of cellular structures is secured by accumulation of protective proteins and quenching of overproduced reactive oxygen species (ROS) [46]. Here, the green seeds showed distinctly lower vigor in comparison to the yellow ones, as can be seen from delayed germination and higher percentage of abnormal seedlings observed on the third day (*t*-test: *p* = 0.02, Figure 1B,C). To a large extent, it can be explained by a significantly lower status of their antioxidant system (Figure 1E,F), whereas residual chlorophylls could underlie accelerated formation of ROS [47,48]. Thereby, slightly increased contents of carotenoids (Figure 1A) were not sufficient to quench ROS, which could lead to slightly compromised stability of membrane structures (Figure 1D). 

To address the effects prospectively related to seed residual chlorophylls at the molecular level, we considered the proteins potentially affecting desiccation tolerance, vigor, and longevity. Among them, late embryogenesis abundant (LEA) proteins preserving cell integrity under desiccation stress [49] were one of the most highly represented groups, accounting for nine proteins (Appendix A). Another important feature underlying high vigor is pre-expressed protein synthesis machinery, which allows fast and concerted germination of seeds and development of seedlings [46]. Not less importantly, rapid degradation of storage proteins is a prerequisite for appropriate mobilization of plant resources for growth of a new plant organism [50]. Also, damaged (e.g., oxidized, glycated, lipoxidized, or misfolded) polypeptides need to be degraded to prevent deleterious effects within the cell [51,52]. In agreement with previous reports [35], proteins involved in protein metabolism were represented with t-RNA ligases, translation initiation, elongation factors, chaperones, and degradation enzymes.

Some enzymes related to protein metabolism showed differential expression in yellow- and green-seeded cultivars. Thus, totally 33 nonredundant t-RNA ligases were identified in yellow pea seed embryos, while 6 unique proteins were additionally annotated in green ones. Translation initiation factors demonstrated similar differences (Appendix A). As a decrease in abundance of translation factors is known to accompany age-related loss of seed vigor [53,54], the observed expression profiles might indicate a higher amenability of green seed embryos to such a loss during a prolonged dormancy period. Protein folding was represented by more than 20 entries (HSP70 and 90, GroEL, T-complex protein TCP-1/cpn60, Appendix A, bin 23). Indeed, HSP70 and GroEL were responsive to osmopriming in Arabidopsis and alfalfa seeds, respectively [55,56], whereas abundance of TCP-1 was decreased in aged alfalfa seeds [57]. Remarkably, several peptidases and DnaJ heat shock protein were identified only in Millennium seeds (Appendix A), facilitating faster degradation of damaged polypeptides and faster mobilization of storage proteins. Interestingly, pea seeds contained acylamino-acid-releasing enzyme-like protein, involved in the degradation of oxidized and glycated proteins [58]. Thus, together with the glyoxalase system and ribulosamine/erythrulosamine 3-kinase, it might contribute to the plant antiglycative machinery.

One of the critical factors affecting vigor, longevity, and, in general, viability of dormant and germinating seeds is the abundance of ROS in seed tissues [59,60]. In agreement with this, a rich pattern of redox proteins (thioredoxin reductase, glutathione peroxidase, monodehydroascorbate reductase, superoxide dismutase, dehydroascorbate reductase, and catalase), earlier identified in other species [32,39,61], was confirmed here (Appendix A). Yellow seed embryos contained eight unique oxidoreductases that might indicate higher levels of oxidative catabolism and higher capacity of electron transfer chains [47], and could potentially increase vigor of yellow seeds.

Deeper analysis of metabolic pathways potentially involved in seed longevity [35,62] (Supplementary information 7) revealed S-adenosylmethionine (SAM) synthase 2 and glycolytic enzyme hexokinase 1 and E1 alpha subunit of the pyruvate dehydrogenase complex. As was earlier shown in priming and ageing experiments [53], SAM synthase is critical for seed vigor. Conversely, enzymes of primary metabolism were shown to be affected by controlled deterioration (accelerated ageing) of seeds (i.e., age-related loss of vigor [63,64]). In this case, the presence of additional isoforms of hexokinase and pyruvate dehydrogenase might not only affect the rate of glycolysis, but also increase production of Ac-KoA and its involvement in the TCA cycle. Remarkably, yellow seeds express three unique aldehyde dehydrogenases-enzymes, reducing reactive carbonyl products of lipid peroxidation and glycation to corresponding alcohols, thereby protecting proteins from oxidative damage [65]. 

To detect protein–protein interaction networks specific to green seed embryos, we performed String database analysis with the proteins found solely in Gloriosa seeds (Figure 8). The largest functional/structural cluster (#1, Figure 8A) was represented by enzymes, involved in nucleotide metabolism and nucleotide-dependent regulatory pathways, as well as protein biosynthesis in chloroplasts and malonyl-CoA biosynthesis (Figure 8B). These functional relations can be explained by simultaneous synthesis of lipids and integral proteins of thylakoid membranes, triggered by activation of phytosulfokine receptors and mediated by calmodulin-related signaling [66]. The second major cluster (#5) comprised proteins related to m-RNA processing and initiation of translation. As can be seen from Figure 8A, it is closely related to the previous one, suggesting an impact on activation of gene expression during growth [67]. Analysis of the major coexpression clusters (Figure 8C, clusters 3 and 5) was in agreement with these results and confirmed activation of protein synthesis (mostly translation and RNA processing directly related to it, Figure 8D) as one of the main features of green-colored seeds (Figure 8A and Appendix A). Further, the coexpression clusters 4 and 7 confirmed the functional annotation data, indicating high representation of photosynthesis-related proteins in the seed proteome (Figure 5).

## 4. Materials and Methods

### 4.1. Reagents and Plant Material

Unless stated otherwise, materials were obtained from the following manufacturers. Carl Roth GmbH & Co (Karlsruhe, Germany): acetonitrile (≥99.95%, LC-MS grade), ethanol(≥99.8%), sodium dodecyl sulfate (SDS) (>99%), tris-(2-carboxyethyl)-phosphine hydrochloride (TCEP, ≥98%); PanReac AppliChem (Darmstadt, Germany): acrylamide (2K Standard Grade), glycerol (ACS grade); AMRESCO LLC (Fountain Parkway Solon, OH, USA): ammonium persulfate (ACS grade), glycine (biotechnology grade), *N*,*N*′-methylene-bis-acrylamide (ultrapure grade), tris(hydroxymethyl)aminomethane (tris, ultrapure grade), urea (ultrapure grade), ammonium bicarbonate (puriss, p.a.); Bioanalytical Technologies 3M Company (St. Paul, MN, USA): Empore™ solid phase octodecyl extraction discs; Component-Reactiv (Moscow, Russia): phosphoric acid (p.a.); Reachem (Moscow, Russia): hydrochloric acid (p.a.), isopropanol (reagent grade), potassium chloride (reagent grade); SERVA Electrophoresis GmbH (Heidelberg, Germany): Coomassie Brilliant Blue G-250, 2-mercaptoethanol (research grade), trypsin NB (sequencing grade, modified from porcine pancreas); Thermo Fisher Scientific (Waltham, MA, USA): PierceTM Unstained Protein Molecular Weight Marker #26610 (14.4–116.0 kDa), PageRullerTM Plus Prestained Protein Ladder #26620 (10–250 kDa); Dichrom GmbH (Marl, Germany): Progenta™ anionic acid labile surfactant II (AALS II) and adaptors for stage-tips; Vekton (St. Petersburg, Russia): acetonitrile (HPLC grade), sucrose (ACS grade), conc. HCl (puriss). All other chemicals were purchased from Sigma-Aldrich Chemie GmbH (Taufkirchen, Germany). Water was purified in-house (resistance 5–15 mΩ/cm) on water conditioning and purification systems Elix 3 UV (Millipore, Moscow, Russia) or Millipore Milli-Q Gradient A10 system (resistance 5–15 mΩ/cm, Merck Millipore, Darmstadt, Germany). 

Pea seeds of the yellow-seeded cultivar Millennium (Appendix A) were obtained from the Research and Practical Center of National Academy of Science of the Republic of Belarus for Arable Farming (Zhodino, Belarus, harvested in the year 2015). The seeds of the green-seeded cultivar Gloriosa (Appendix A) were provided by the Institute of Vegetable Growing, National Academy of Science (Minsk, Belarus, harvested in the year 2015).

### 4.2. Analysis of Physiological and Biochemical Parameters of Seed Quality

The seeds were germinated on filter paper in a thermostat at 22 °C. To address the kinetics of germination, the numbers of germinated seeds were monitored for 10 days on a daily basis, before the seeds were classified as (i) nongerminating, and those producing (ii) normal and (iii) abnormal seedlings upon germination, as defined by the International Seed Testing Agency (ISTA) [68]. Electrical conductivity test relied on the method of Matthews et al. [69]. For this, seeds (*n* = 5) were incubated in 25 mL of distilled water for 2 h at 22 °C. The conductivity of the aqueous medium was determined with a conductometer HI 8733 (HANNA Instruments Deutschland GmbH, Vöhringen, Germany) before and after the incubation. Determination of photosynthetic pigments (*n* = 3) relied on the method of Lichtenthaler and Wellburn [70] (for details see Appendix A). Hydrogen peroxide was quantified by the method of Bilova et al. [71], whereas the contents of lipid peroxidation products were determined as malondialdehyde (MDA) equivalents, as described by Frolov and coworkers [26] (see Protocols S1-2 and S1-3, respectively). 

### 4.3. Protein Isolation 

Pea seeds (10 per biological replicate, *n* = 3) were frozen in liquid nitrogen and ground in a Mixer Mill MM 400 ball mill with a Ø 20 mm stainless steel ball (Retsch, Haan, Germany) at a vibration frequency of 30 Hz for 2 × 1 min. The resulting ground material (approximately 50 mg per replicate) was placed in 2 mL safe-lock polypropylene tubes and stored at −80 °C prior to protein extraction, which relied on the method of Frolov and coworkers [16] with some modifications. Briefly, the tubes with plant material were transferred on ice, and 1 min later supplemented with 700 µL of cold (4 °C) phenol extraction buffer, containing 0.7 mol/L sucrose, 0.1 mol/L KCl, 5 mmol/L ethylenediaminetetraacetic acid (EDTA), 2% (*v*/*v*) mercaptoethanol and 1 mmol/L phenylmethylsulfonyl fluoride (PMSF) in 0.5 mol/L tris-HCl buffer (pH 7.5). The suspensions were vortexed for 30 s, before 700 µL of cold phenol (4 °C) saturated with 0.5 mol/L tris-HCl buffer (pH 7.5) were added. After further mixing for 30 s, the samples were shaken for 30 min at 900 rpm (4 °C) and centrifuged (5000× *g*, 30 min, 4 °C). Afterwards, the phenolic (upper) phases were transferred to new 1.5 mL polypropylene tubes, and washed two times with equal volumes of the phenol extraction buffer (after each buffer addition: vortexing 30 s, shaking for 30 min at 900 rpm at 4 °C, and centrifugation at 5000× *g* for 15 min at 4 °C). Then, the supernatants were transferred to 1.5 mL polypropylene tubes, and proteins were precipitated by addition of a fivefold volume of ice-cold 0.1 mol/L ammonium acetate in methanol, followed by storage overnight at −20 °C. Next morning, the protein fraction was collected by centrifugation (10 min, 5000× *g*, 4 °C), and the supernatants were discarded. The pellets were washed twice with two volumes of methanol (relative to the volume of the phenol phase), and twice with the same volume of acetone (both at 4 °C). Each time, the samples were centrifuged (5000× *g*, 10 min, 4 °C) after resuspension. Finally, the cleaned pellets were dried under air flow in a fume hood for 1 h, reconstituted in 100 µL of shotgun buffer (8 mol/L urea, 2 mol/L thiourea, 0.15% AALS II in 100 mmol/L tris-HCl, pH 7.5), and protein contents were determined by 2-D Quant Kit (GE Healthcare, Taufkirchen, Germany) according to Matamoros and coworkers [41] or by the Bradford method as described by Frolov and coworkers [72]. The precision of the assay was verified by SDS-PAGE according Greifenhagen et al. [73] with minor modifications (for details see Protocol S1-4). 

### 4.4. Depletion of Seed Storage Proteins by Protamine Sulfate and Tryptic Digestion

Depletion of seed storage proteins relied on the protamine sulfate precipitation procedure (PSP) of Kim et al. [9] with some modifications. In detail, 95 µL of the total protein fractions, isolated from green and yellow seeds (*n* = 3) as described above, were supplemented with 855 µL of 0.077% (*w/v*) protamine sulfate in 20 mmol/L MgCl_2_/0.5 mol/L Tris-HCl (pH 8.3) to yield a final protamine sulfate concentration of 0.07% (*w*/*v*). After centrifugation (12,000× *g*, 10 min, 4 °C), supernatants were transferred to 5 mL polypropylene tubes and supplemented with four volumes of 12.5% trichloroacetic acid (TCA) in cold (−20 °C) acetone and left at −20 °C overnight. Next morning, the pellets were washed three times with 1 mL of cold acetone by sequential resuspending and centrifugation (12,000× *g*, 10 min, 4 °C), and clean pellets were dried in a fume hood for 1 h. The dried pellets were reconstituted in 100 µL of shotgun buffer containing 0.15% AALS, and the protein contents were determined by 2-D Quant Kit without further dilution. The digestion procedure relied on the protocol of Frolov and coworkers [26] with minor modifications (See Protocol S1-5 for details).

### 4.5. NanoHPLC-ESI-Q-Orbitrap Analysis

Dried tryptic digests were reconstituted in 100 µL of aq. acetonitrile (final concentration of 3% *v/v*) containing 0.1% (*v*/*v*) aq. formic acid, and 1 µL (250 ng) of peptide mixture was loaded onto an Acclaim PepMap 100 C18 trap column (75 µm × 20 mm, nano-viper, 3 µm particle size). After trapping, the peptides were separated on a PepMap™ RSL C18 column (75 µm × 50 cm, 2 µm particle size) using an EASY-nLC 1000HPLC system coupled on-line to a Q-Exactive Plus mass spectrometer via an EASY-spray nano ion source (all Thermo Fisher Scientific, Bremen, Germany). The details of the chromatographic method are summarized in Appendix A. The nano-LC-Orbitrap-MS analysis relied on data-dependent acquisition (DDA) experiments performed in the positive ion mode, comprising a survey Orbitrap-MS scan and dependent MS/MS scans for the most abundant signals in the following 5 s (at certain tR) with charge states ranging from 2 to 6. The mass spectrometer settings and DDA parameters are summarized in Appendix A. The database search relied on Proteome Discoverer 2.2 software (Thermo Fisher Scientific, Bremen, Germany), SEQUEST search engine, and a combined sequence database comprising proteomes of *Medicago truncatula* Gaertn, *Lotus japonicus* (Regel) K. Larsen, and *Phaseolus vulgaris* L [41] with addition of protein sequences, prospectively related to seed longevity and color (see Appendix A for details). The redundancy of the database was eliminated by the СD-HIT algorithm [74] with the sequence identity cutoff set to 1. The database search parameters are summarized in Appendix A.

### 4.6. Data Analysis and Postprocessing

Functional annotation of pea proteins was done withVisual Basic for Applications (VBA). The corresponding script (Appendix A) was run within Excel instance and managed automatic extraction, appendment, and rearrangement of tabular data, organized in spread sheets. Further, it established access to a laboratory web server via a simplified HTTP protocol. The server, in turn, was run under Apache 2.4, used as a CGI front end for a set of bash scripts. These scripts managed previously cached data, performed actual queries to external public databases, or ran locally installed programs. After screening against UniProt [75] or KEGG [76] databases, amino acid sequences, gene ontology (GO) identifiers, and KEGG Orthology (KO) terms with text annotations were extracted as text files [77]. Annotation relied on the functional bins used by the MapMan software suite [78]. Cellular localization was predicted by the NGLOC algorithm [79]. Finally, all annotations were fitted back into the original Excel spreadsheet and clustered by localization terms or by top MapMan hierarchy levels, before the data was manually inspected, corrected, or further automatically processed. 

Analysis of protein networks relied on a search against the *Arabidopsis thaliana* proteome in the String database (for details see Protocol S1-6). Only highly confident protein entries with interaction scores ≥0.7 were exported for building protein interaction maps based on (i) experimentally confirmed evidences and (ii) coexpression. Venn diagrams were constructed using the web-based tool InteractiVenn [80] based on the information given in Proteome Discoverer 2.2 output in columns “Found in Samples”.

## 5. Conclusions

The quality of plant seeds is a prerequisite for sustainable agriculture and food production. As legume crops represent the main source of food protein worldwide, the quality of their seeds requires special attention. Using comprehensive methods of LC-MS-based proteomics, we succeeded in extensively covering the pea seed proteome and assigning functions to individual nonredundant proteins. This systematic approach allowed the identification of characteristic metabolic features of green-coloured seeds in comparison to yellow-coloured ones. Thereby, alterations in the proteome of green seeds, prospectively related to their reduced vigor and longevity, were revealed.

## Figures and Tables

**Figure 1 ijms-19-04066-f001:**
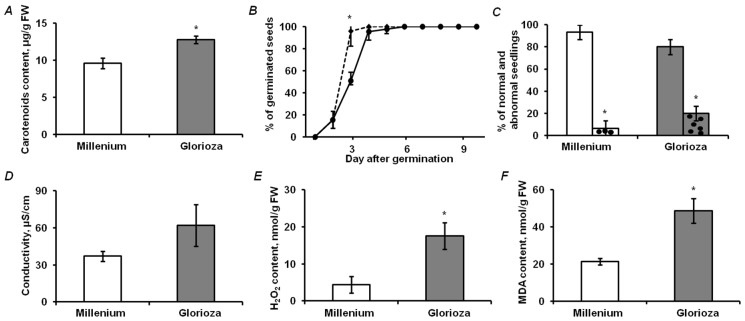
Physiological and biochemical parameters of seed quality, acquired for the mature seeds of cultivar Millennium (solid line, white columns) and Gloriosa (dashed line, grey columns): (**A**) contents of carotenoids, (**B**) kinetics of germination, (**C**) distribution of seedlings by morphology (filled and dotted columns indicate normally and abnormally developed seedlings, respectively), (**D**) electrolyte leakage, expressed as medium conductivity, µS/cm, as well as (**E**) tissue levels of hydrogen peroxide and (**F**) lipid peroxidation products, expressed as malondialdehyde (MDA) equivalents. Asterisks denote statistically significant difference between cultivars (**A**,**B**,**E**,**F**) or indicate difference between percentage of nongerminated seeds in two cultivars (**C**), *t*-test: *p* < 0.05.

**Figure 2 ijms-19-04066-f002:**
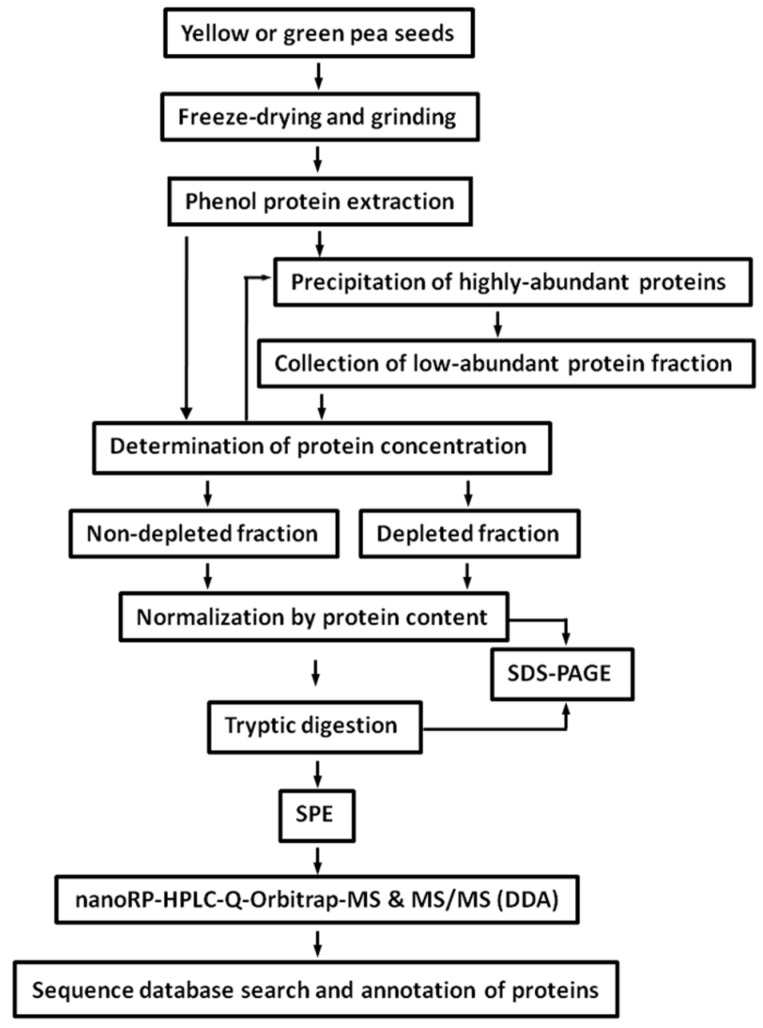
Experimental workflow, employed for characterization of the pea seed proteome.

**Figure 3 ijms-19-04066-f003:**
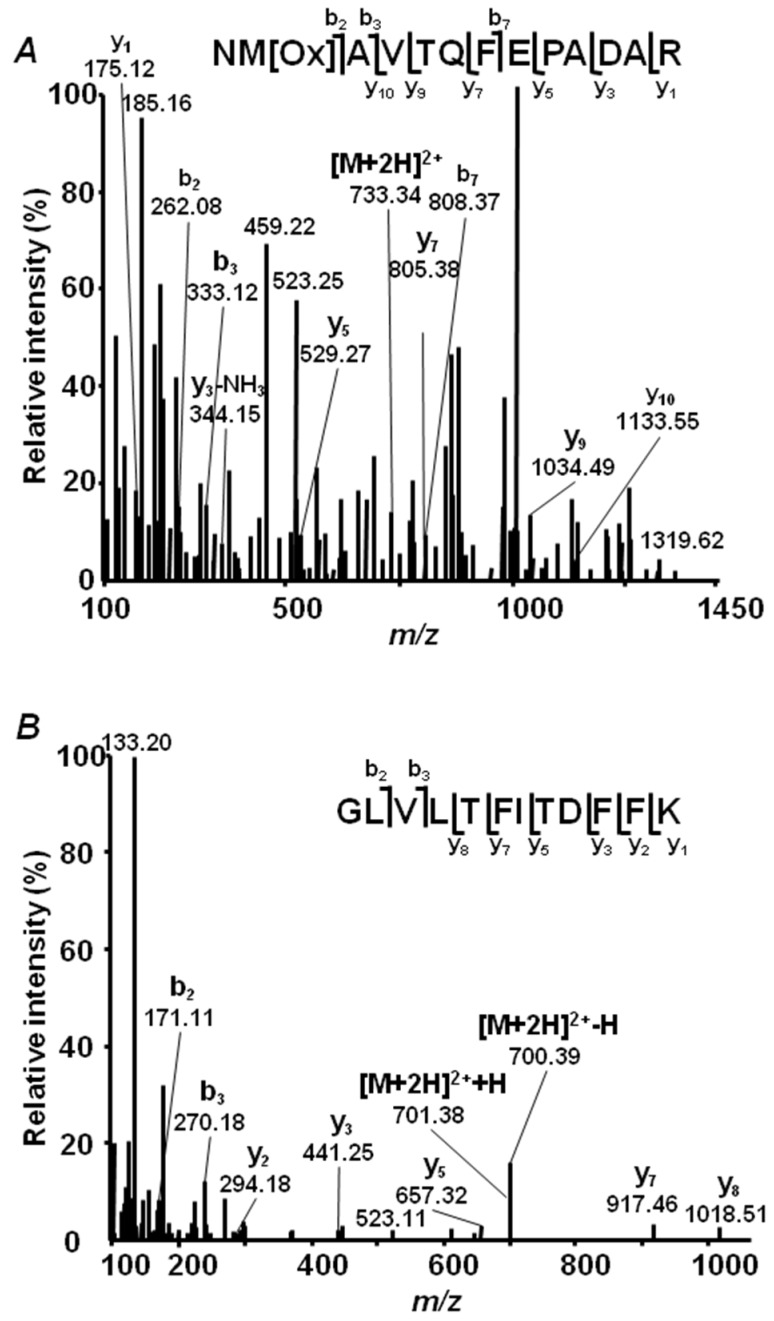
Tandem mass spectra of (**A**) *m*/*z* 733.3 corresponding to the peptide NM_Ox_AVTQFEPADAR, which represents residues 131–143 of aminopeptidase (protein accession Lj1g3v1787580.1, *Lotus japonicus*), (**B**) *m*/*z* 700.89 corresponding to the peptide GLVLTFITDFFK, which represents residues 179–190 of the eukaryotic translation initiation factor (protein accession G7KRJ1, *Medicago truncatula*), and (**C**) *m*/*z* 510.79 corresponding to the peptide SVAGEIFGLK, which represents residues 170–180 of the protein from glutamine synthetase family (protein accessionV7BHN3, *Phaseolus vulgaris*).

**Figure 4 ijms-19-04066-f004:**
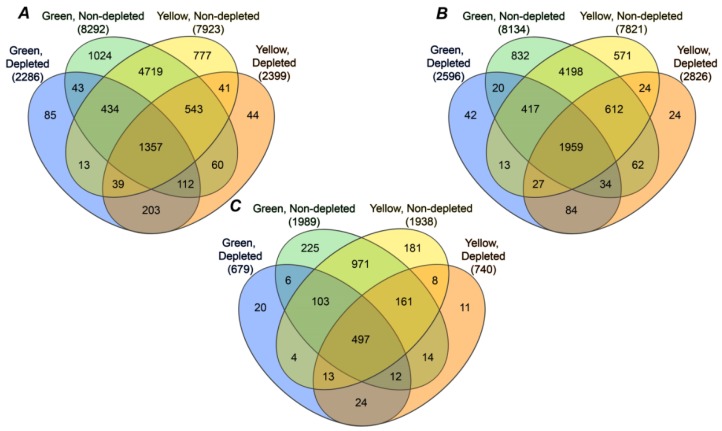
The numbers of tryptic (**A**) peptides, (**B**) proteins, and (**C**) protein groups, identified in yellow-colored (cultivar Millennium) and green-colored (cultivar Gloriosa) seeds with and without depletion of highly abundant proteins with protamine sulfate.

**Figure 5 ijms-19-04066-f005:**
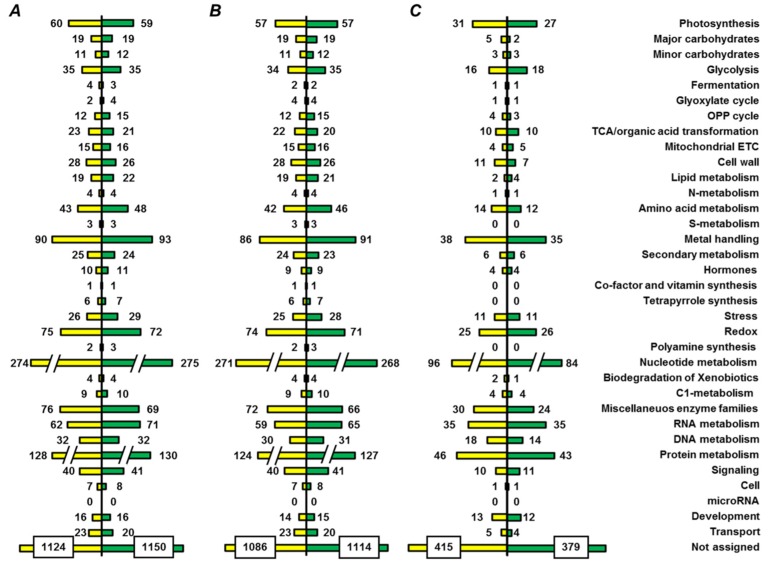
Functional annotation of proteins, identified in yellow seeds of cultivar Millennium (yellow color) and in green seeds of cultivar Gloriosa (green color), (**A**) both in nondepleted samples and after treatment with 0.07% (*w/v*) protamine sulfate, (**B**) only in nondepleted fraction and (**C**) protamine-sulfate supernatant fraction.

**Figure 6 ijms-19-04066-f006:**
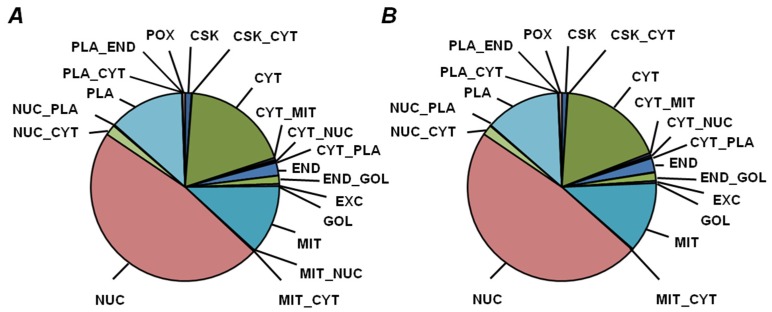
Prediction of subcellular localization of proteins, identified (**A**) in yellow seeds of cultivar Millennium and (**B**) in green seeds of cultivar Gloriosa, both in nondepleted samples and after treatment with 0.07% (*w/v*) protamine sulfate. Subcellular fractions: CSK, cytoskeleton; CYT, cytoplasm; END, endoplasmic reticulum; EXC, extracellular/secreted; GOL, Golgi apparatus; MIT, mitochondria; NUC, nuclear; PLA, plasma membrane; POX, peroxisome; doubled labels denote corresponding combinations of possible localization.

**Figure 7 ijms-19-04066-f007:**
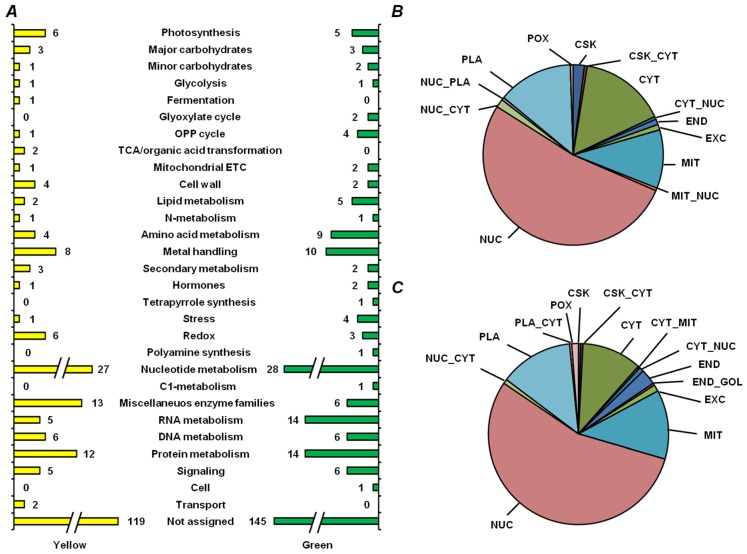
Functional annotation (**A**) and prediction of subcellular localization (**B**,**C**) of proteins, unique for yellow seeds of cultivar Millennium (yellow color in panel **A** and **C**) and for green seeds of cultivar Gloriosa (green color in panel **A** and **B**), both in nondepleted samples and after treatment with 0.07% (*w/v*) protamine sulfate. Subcellular fractions: CSK, cytoskeleton; CYT, cytoplasm; END, endoplasmic reticulum; EXC, extracellular/secreted; GOL, Golgi apparatus; MIT, mitochondria; NUC, nuclear; PLA, plasma membrane; POX, peroxisome; doubled labels denote corresponding combinations of possible localization.

**Figure 8 ijms-19-04066-f008:**
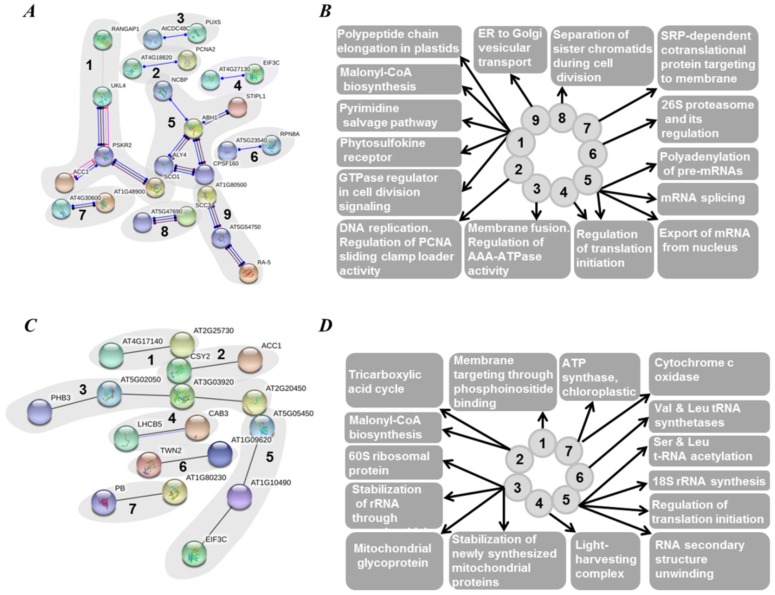
String database analysis (https://string-db.org/) of interaction networks formed by proteins, unique for green seeds (cultivar Gloriosa). The networks (**A**,**C**) and related functional pathways (**B**,**D**) relied on experimentally derived functional and/or structural evidences (**A**,**B**) and coexpression data (**C**,**D**) with interaction score ≥ 0.7 (corresponded to highly confident entries). Color coding of interactions: black, reaction; blue, binding; light blue, phenotype; green, activation; purple, catalysis; red, inhibition; pink, posttranslational modifications (PTMs); yellow, transcriptional regulation. Filled and empty nodes denote proteins with known and unknown structure, respectively. Detailed description of protein data is provided in Appendix A.

**Figure 9 ijms-19-04066-f009:**
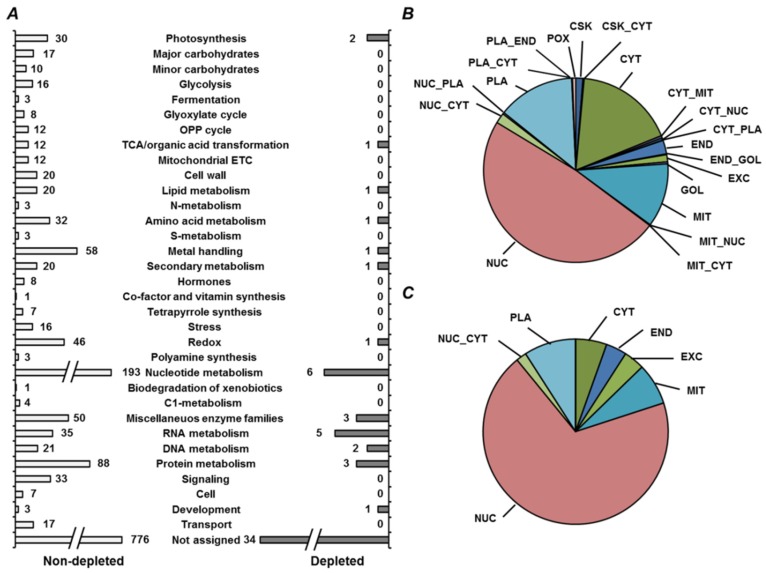
Functional annotation (**A**) and prediction of subcellular localization (**B**,**C**) of proteins, identified both in yellow seeds of cultivar Millennium and in green seeds of cultivar Gloriosa in nondepleted samples (white color in panels **A** and **B**) and after treatment with 0.07% (*w/v*) protamine sulfate (grey color in panels **A** and **C**). Subcellular fractions: CSK, cytoskeleton; CYT, cytoplasm; END, endoplasmic reticulum; EXC, extracellular/secreted; GOL, Golgi apparatus; MIT, mitochondria; NUC, nuclear; PLA, plasma membrane; POX, peroxisome; doubled labels denote corresponding combinations of possible localization.

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
