# Peer review of "Proteome Map of Pea (Pisum sativum L.) Embryos Containing Different Amounts of Residual Chlorophylls"

_ijms, 2018, doi:10.3390/ijms19124066_

Reviewer 1 Report

The MS can be published in this Journal. Please check the address of authors, especially 5.

Author Response

We thank Reviewer for appreciating the manuscript.

The addresses of all authors have been checked.

Reviewer 2 Report

The submitted manuscript entitled “Proteome map of pea (Pisum sativum L.) embryos containing different amounts of residual chlorophylls” presented the methodology for proteomic analyses of seeds of pea. The authors annotated the functions and intracellular localization of the proteins comprehensively. The authors also identified proteins that differentially expressed in yellow and green seeds of pea. This manuscript is within the scope of International Journal of Molecular Sciences and I believe that this manuscript includes important information regarding proteomics in pea. However, I would request the authors to revise this manuscript for reconsideration.

 This manuscript may have problems in presentation. The key message of this study seems to be unclear. Large parts of the manuscript describe about methodology of proteomic analyses of pea seeds, and the reviewer believes that the highest value of this manuscript is in the methodology. However, from the title, abstract, and introduction, it seems like that the purpose of the study was identification of proteins differentially expressing in yellow seeds and green seeds through the proteomic analyses. Further in discussion, the authors associated the results of proteomic analyses with tolerance of seeds to desiccation. The story by the authors may be hard to understand for readers. I understood that the authors discussed tolerance of seeds to desiccation on the basis of the data shown in Figure 1 presenting the different duration of germination and the different percentages of normal seedlings between yellow seeds and green seeds. However, no evidences were shown that the different “vigor” was owing to different tolerance to desiccation between yellow seeds and green seeds. Other factors can affect “vigor” of seed. Furthermore, the differences of expressed proteins identified by the proteomic analyses reflect all the differences between 2 cultivars subjected to the analyses. When you compare materials with different color or different desiccation tolerance having similar genetic background, such a discussion may be acceptable. My suggestion to authors is to reconstruction of the story of the manuscript. The key message of this manuscript should be “an efficient protocol for proteomic analyses in pea” or something like that. Comparison of proteins between yellow and green cultivars should be presented as a test case of the proteomic analyses. To me, linking the results of proteomic analyses to seed color or to tolerance to desiccation seems beyond this study at this moment.

 Other points were listed below:

 Line 19: The authors described “Obviously, changes in seed protein patterns might directly affect both of these aspects.” If it is “obvious”, “might” should be removed.

 Line 22: The authors described about color of seed of pea, yellow and green, here. It sounds to be very sudden. A lead sentence showing what the authors were trying to discover through the comparison of the protein between yellow and green pea seed would improve readability of the abstract.

 Line 69: What does “higher redox status” mean? Higher oxidative status? I had a glance through the cited paper numbered 19. The paper certainly mentioned about relationship between seed color and stress tolerance, however, the paper did not reveal about seed of pea. This paper may be a very weak basis for this study. 

 Line 39: I would remove “might.”

 Line 79: A short leading sentence showing what the authors were trying to discover by their analyses should be added to improve readability. For example, “to identify the differences of physiological characteristics between yellow seed and green seed, we conducted...” 

 Line 85: The purposes of measurements of conductivity, lipid peroxidation products, and H2O2 should be explained briefly.

 Figure 1: It was not clear what the asterisks on Figure 1c implied. It seems like the percentages of normal seedlings and abnormal seedlings in each cultivar were significantly different. However, it obviously makes no sense. Comparison of the percentages of normal seedlings between the cultivars may be required.

 Lane 104: “Among lanes” instead of “between lanes”?

Author Response

We thank Reviewer for the thoughtful review and highly appreciate the valuable comments and suggestions to improve the manuscript. Following these advices we performed all required changes in corresponding sections, as indicated in the following rebuttal addressing each aspect.

Reviewer 3 Report

Dear Authors,

Reviewer comments ijms-394283

The manuscript entitled „Proteome map of pea (Pisum sativum L.) embryos containing different amounts of residual chlorophylls“ represents a useful study aimed at an investigation and a comparison of proteome composition in green-seeded and yellow-seeded pea cultivars using the plain protein extracts as well as extracts depleted from highly abundant proteins. The manuscript provides very useful data on proteome composition in the two pea cultivars differing in chlorophyll content in their seeds.

I have only a few comments on the manuscript.

1/ I suggest to add a scheme comparing major differences in the seed proteome composition between green-seeded pea and yellow-seeded pea plants as Figure 10.

In Figure 9A, there are only depleted and non-depleted proteins which according to the figure legend, have to belong to both pea cultivars Millenium and Gloriosa. I do not understand how these two cultivars are distinguished.

2/ Materials and methods, part Data analysis and post-processing: The authors have to specify the version of protein databases used including date of download, taxonomy, and search criteria (lines 509-510: „After screening against UniProt or KEGG databases, amino acid sequences, gene ontology (GO) identifiers and KEGG Orthology (KO) terms with text annotations were extracted as text files…“

3/ Formal comments:

In Figure 2 showing experimental workflow employed for characterization of the pea seed proteome, correct the typing error in the term „Precipitation of highly-abundant proteins“ (NOT: „Pecipitation“).

In Results, line 222: Modify the sentence as follows: „For this aim,…“ or „For this purpose,…“ (NOT: „For this,…“).

Results, line 235: Change the position of commas in the sentence as follows: „Accordingly, the supernatants obtained after supplementation with protamine sulfate were dominated by nuclear proteins….“

Discussion, line 252: Add a comma preceding the word „generally“ in the sentence „…and, generally, in contrast to most of the other published techniques,…“

Discussion, line 253: Add a comma following the words „Indeed, in our hands,…“

Discussion, line 270: Add the words „are available“ at the end of the sentence „To date, studies of the legume seed proteome mostly relied on gel-based techniques (typically 2D-GE and MALDI-TOF/TOF-MS) yielding up to several hundreds of protein identifications are available.

Discussion, line 276: Add a comma following the word „However,…“

Discussion, line 277: Add the word „fact“ following the words „The possible reasons for this fact might be…“

Discussion, line 351: Remove a comma in the sentence „…proteins involved in protein metabolism were represented with t-RNA ligases, translation initiation, elongation factors, chaperones, and degradation enzymes.“

Discussion, line 367: Add a comma following the words „Thus, together with glyoxalase systém and ribulosamine/erythrulosamine 3-kinase,…“

Discussion, line 393: Correct the typing error in the term „calmodulin-related signaling“ (NOT „calmoduline-related signaling“).

4/ Supplementary materials: In Supplementary materials, all identified unique peptides matched to each identified protein have to be provided including peptide sequences and appropriate statistics (FDR values).

 Final recommendation: Accept after a minor revision.

 Author Response

We thank Reviewer for the thoughtful review and highly appreciate the valuable comments and suggestions to improve the manuscript. Following these advices we performed all required changes in corresponding sections, as indicated in the following rebuttal addressing each aspect.

Round  2

Reviewer 2 Report

I believe that the resubmitted manuscript entitled “Proteome map of pea (Pisum sativum L.) embryos containing different amounts of residual chlorophylls” has been well-revised. I appreciate the authors for sincere responses to the reviewer's comments. I think that the aims of the study have been more clarified and readability has been improved so much by the modifications. Furthermore, the letter from the authors made me realize my misunderstanding: desiccation tolerance is not focal point of the paper but is assumed characteristics of yellow seeds of pea through some physiological experiments and the proteome analyses by the authors. To me, this version is suitable for publication in the IJMS.